# ML-based detection of depressive profile through voice analysis in WhatsApp™ audio messages of Brazilian Portuguese Speakers

Victor H. O. Otani[1,2], Felipe O. Aguiar[1,2], Thiago P. Justino[3], Hudson S. Buck[4], Luiza B. Grilo[1], Matheus F. Figueiredo[1], Pedro M. Uchida[5], Daniel A. C. Vasques[2], Thaís Z. S. Otani[1,2], João Ricardo N. Vissoci[6], Lucas M. Marques[1,2*], Ricardo R. Uchida[1,2]

**1** Mental Health Department, Santa Casa de São Paulo School of Medical Sciences, São Paulo, São Paulo, Brazil, **2** Artificial Intelligence Research Division, Infinity Doctor's Inc., São Paulo, Brazil, **3** Faculty of Health Sciences, Federal University of Grande Dourados (UFGD), Dourados, Mato Grosso do Sul, Brazil, **4** Eólas Applied Neuroscience, Mogi das Cruzes, São Paulo, Brazil, **5** Pontifical Catholic University of Campinas, Campinas, São Paulo, Brazil, **6** Division of Emergency Medicine, Department of Surgery, Duke Global Health Institute, Duke University, Durham, North Carolina, United States of America,

* lucasmurrins@gmail.com

## Abstract

Depression is a prevalent mental health condition that significantly impacts individuals' daily lives, work productivity, relationships, and overall well-being. The lack of reliable biomarkers complicates screening, contributing to underdiagnosis. Depression's impact on voice and acoustic parameters enables differentiation between adaptive and non-adaptive mood profiles, offering potential classifiers for screening. This study evaluates the capability of seven distinct machine learning models to identify depression in speech samples. WhatsApp™ audio messages (WA), clinical, and sociodemographic data were collected from 160 individuals divided into two groups: one for algorithm development and the other for testing. Each group included patients with Major Depressive Disorder and healthy controls. In the test group, participants were interviewed using the Mini-International Neuropsychiatric Interview (MINI), and their WhatsApp™ audio recordings included both structured and semi-structured formats. After pre-processing the audio, 68 acoustic features were used to train the machine learning models. Results shows that: i) The algorithms evaluated WhatsApp™ audio recordings from the test group, achieving peak accuracies of 91.67% for women and 80% for men, with an AUC of 91.9% for women and 78.33% for men. ii) The accuracy of Machine Learning (ML) classification varies depending on the type of audio instruction provided. ML can classify, with reasonable accuracy, whether a WhatsApp™ audio message represents a depressive patient or a healthy individual. Future studies should further explore the relationship between voice characteristics, different mood profiles, and emotional states.

**Data availability statement:** The dataset cannot be publicly shared due to ethical restrictions imposed by the Human Research Ethics Committee of Santa Casa de São Paulo. Data requests may be directed to: Email: cepsc@santacasasp.org.br Address: Rua Dr. Cesário Mota Junior, 112, 1º andar, Sala 1, Hospital Central, São Paulo, SP, Brazil Phone: +55 11 2176-7000 ext. 7711.

**Funding:** This work was financially supported by Infinity Doctors Inc. for the following authors: RU, FA, DV, TO, LM, and VO. The funder provided financial support for the time and effort of these authors during the development of this study. The funders had no role in study design, data collection and analysis, decision to publish, or preparation of the manuscript.

**Competing interests:** he authors have read the journal's policy and the authors of this manuscript have the following competing interests: RU, FA, DV, TO, LM, and VO report employment and equity or stock ownership in Infinity Doctors Inc. LM and JRV are members of the Advisory Board of PLOS Mental Health. The authors would like to declare the following patents/patent applications associated with this research: RU holds a pending patent (#TPP77348) related to this work submitted as a Provisional Patent Application. This does not alter our adherence to PLOS policies on sharing data and materials.

## Introduction

Depression is a widespread mental health condition, affecting approximately 280 million people globally and accounting for 5% of the adult population, according to estimates from the World Health Organization (WHO) in 2019 [1]. The condition is significantly more prevalent among women, who are nearly twice as likely as men to experience depression [2]. Depression can profoundly impair various aspects of life, including academic performance, work productivity, and social relationships, while also increasing the risk of several health conditions such as suicide, cardiovascular diseases, cancer, diabetes, and respiratory diseases [3–5]. In Brazil, the prevalence of depression is notably high, with 5.8% of the population affected, the highest rate in Latin America [6].

Suicide remains one of the leading causes of mortality worldwide, particularly among young individuals, and is strongly associated with depressive symptoms and suicidal ideation. Approximately 15% of individuals who experience Major Depressive Disorder (MDD) attempt suicide at some point in their lives [7]. Early detection and timely intervention are therefore critical components of suicide prevention and mental health promotion.

Digital technologies offer a unique opportunity to improve accessibility to mental health screening. Among them, WhatsApp™, one of the most widely used instant messaging applications globally, represents a promising public health tool for detecting depressive symptoms through passive and non-invasive voice analysis. Such an approach could enable large-scale, low-cost, and early identification of depressive states, supporting both clinical care and population-level monitoring.

Beyond its clinical implications, the implementation of machine learning–based tools for depression detection carries substantial societal and economic potential. By facilitating early identification, these technologies may help reduce the burden of untreated depression, stimulate help-seeking behaviors, and improve healthcare efficiency through optimized resource allocation. In turn, this could promote a paradigm shift toward preventive and personalized mental health care, while reducing the overall economic costs associated with depression and suicide.

The heterogeneity of depression, with its various subtypes and clinical presentations, complicates diagnosis. Current diagnostic approaches rely heavily on clinical evaluations and psychometric instruments such as the Beck Depression Inventory (BDI), Hamilton Depression Rating Scale (HAM-D), and Montgomery-Åsberg Depression Rating Scale (MADRS) [8–10]. Although these tools provide valuable insights into the presence and severity of depression, they often lack the specificity required for clinical phenotyping and individualized treatment strategies [11].

Moreover, broad screening tools such as the Mini-International Neuropsychiatric Interview (MINI) offer comprehensive diagnostic evaluations but are time-consuming and require specialized personnel, limiting their scalability in large-scale or task-sharing contexts [12]. Despite significant advances in understanding the pathophysiology and treatment of depression, no definitive complementary tests are available to establish or exclude a diagnosis, posing challenges for early detection by general practitioners and specialists alike [13].

Recent developments in machine learning (ML) and voice signal processing have opened promising avenues for scalable, non-invasive tools for depression detection. However, most existing studies rely on laboratory-based data or scripted tasks, limiting their ecological validity and practical applicability. To our knowledge, this is the first study to apply clinically validated ML models to voice data captured via WhatsApp™—a platform already widely integrated into mental health communication in real-world settings.

Studies have demonstrated the potential of ML models to classify depression using voice biomarkers, with some suggesting that even minor depressive symptoms can be detected through these methods [14,15]. For instance, fusion-based deep learning (FBDL) techniques have shown higher accuracy in classifying depression among elderly individuals compared to non-fusion methods [16], while k-Nearest Neighbors (kNN) models have been particularly effective in diagnosing bipolar disorder, albeit less accurate for depressive disorders [17]. Additionally, decision tree screening models have achieved notable accuracy rates, further underscoring the potential of ML in this field [17].

This study advances the field by combining machine learning with real-world voice recordings and rigorous psychiatric validation. All participants were personally evaluated by trained psychiatrists using DSM-5 and MINI-based structured interviews, ensuring diagnostic accuracy and methodological robustness. Moreover, the two groups used for training and testing were intentionally designed with distinct voice collection protocols (spontaneous versus structured tasks) to evaluate generalizability in practical clinical conditions. The choice to use WhatsApp™ as a collection medium was deliberate: rather than optimizing audio fidelity through studio recordings, the goal was to simulate the conditions under which such tools would be implemented in clinical practice—quick, remote, and platform-based. In this initial phase, we modeled depression as a binary classification problem (Major Depressive Disorder vs. healthy control) to test the feasibility of voice-based machine learning approaches in real-world settings. This design aligns with the study's goal of evaluating screening potential rather than diagnostic granularity. Future extensions will address ordinal or multiclass classification reflecting depression severity (e.g., mild, moderate, severe) and regression-based modeling against validated clinical scales such as the MADRS and PHQ-9.

Given these advancements, exploring ML voice models for depression detection in the Portuguese language is a valuable endeavor. This study hypotheses that ML-driven analysis of vocal signals captured in WhatsApp™ audio messages can accurately differentiate between depressive and non-depressive individuals. The potential impact of this research is substantial, offering a non-invasive, and scalable approach to mental health screening. Such advancements could facilitate early detection, improve diagnostic accuracy, and enhance personalized treatment strategies, ultimately reducing the societal and economic burden of depression [5,13,18]. Implementing these ML models in real-world settings could significantly improve access to mental health care, particularly in underserved populations, and promote timely interventions [19]. In this context, the present study aimed to investigate the capability of ML models to accurately classify depressive profiles using vocal signals captured in WhatsApp™ audio messages (WA) from Brazilian Portuguese speakers.

## Methods and analysis

### Design

Two distinct datasets were collected for this study, each consisting of volunteers diagnosed with and without Major Depressive Disorder (MDD). One dataset was used to develop and internally validate the machine learning models, while the other served as an independent external validation set. This design was intentional, allowing us to assess model generalizability across different diagnostic settings, collection protocols, and task types. Participant recruitment was conducted prospectively between May 1, 2023, and July 31, 2024. All participants were over 18 years old, provided written informed consent, and were recruited in compliance with ethical standards. No minors were included in the study.

## Ethics statement

This study was approved by the Ethics Committee for the Analysis of Research Projects (CEPSC) of the University of Mogi das Cruzes (CAAE: 67711023.1.0000.5497). All participants were over 18 years of age and provided written informed consent prior to participation. No minors were included in the study.

To protect participant privacy and confidentiality, all voice recordings were de-identified immediately after collection by removing any personal identifiers from the associated metadata and replacing participant names with randomly generated alphanumeric codes. Audio files were stored on secure, access-controlled institutional servers with encryption both in transit and at rest. Access to the raw audio data was restricted to the core research team and granted only for purposes directly related to the study, in accordance with the approved research protocol.

Processed datasets used for feature extraction contained no information that could be linked back to individual participants. Data sharing for reproducibility purposes will be performed only in anonymized form, in compliance with applicable regulations and ethical guidelines. All procedures were conducted in accordance with the ethical standards of the institutional and national research committees and with the 1964 Declaration of Helsinki and its later amendments.

## Sample

Two independent data collection procedures were conducted: one for training the machine learning models and another for testing their performance on previously unseen data. This design ensured that the evaluation process was conducted on a fully independent dataset, minimizing the risk of overfitting and enhancing external validity.

The training dataset comprised 86 participants (age: $44 \pm 12$ years), including outpatients clinically diagnosed with Major Depressive Disorder (MDD) according to the Diagnostic and Statistical Manual of Mental Disorders (DSM-5) criteria through structured psychiatric evaluations conducted by experienced psychiatrists in private clinical settings (N = 45; 37 women, 8 men). The control group consisted of healthy volunteers (N = 41; 30 women, 11 men) with no self-reported depressive symptoms or history of psychiatric treatment, recruited through advertisements. Detailed demographic information is provided in Table 1.

The test dataset consisted of 74 participants (age: $35 \pm 14$ years), evaluated in a separate institutional context. Diagnoses were established using the Mini International Neuropsychiatric Interview (MINI; Sheehan et al., 1998), selecting outpatients with MDD and no history of manic, psychotic, or substance-related disorders (N = 33; 17 women, 16 men). Depression severity was measured using the Montgomery-Åsberg Depression Rating Scale (MADRS), with a mean score of $26 \pm 9$. Healthy volunteers (N = 41; 21 women, 20 men) without any mental disorder, as confirmed by the MINI, were included as controls. Details of participants' ages are presented in Table 1.

Diagnostic procedures thus differed between datasets: DSM-5–based clinical evaluations were used in the training dataset, while the MINI interview was applied in the test dataset. Although both instruments are validated and exhibit high concordance for MDD diagnosis, subtle differences in structure, duration, and interviewer expertise may introduce mild heterogeneity in diagnostic labeling. In addition, the two datasets differed in speech content (spontaneous speech for

**Table 1. Ages of major depressive disorder (MDD) and healthy volunteers (HV) groups.**

| Group | Age (year), mean (SD) | MDD | HV | P Value |
|---|---|---|---|---|
| Test | | | | |
| | Male | 39 (14) | 32 (12) | 0.094 |
| | Female | 36 (15) | 35 (14) | 0.793 |
| Training | | | | |
| | Male | 43 (17) | 36 (12) | 0.338 |
| | Female | 47 (11) | 43 (10) | 0.088 |

training vs. semi-structured tasks for testing) and demographic distribution (test sample slightly older, particularly among women). These differences reflect the distinct recruitment contexts and were intentionally preserved to evaluate cross-domain generalization under ecologically valid conditions. Nevertheless, they are explicitly acknowledged in the Discussion and Limitations as potential sources of variability to be harmonized in future data collections.

All participants were screened to exclude potential confounding physiological or neurological factors, such as dementia, cerebrovascular disease, chronic respiratory conditions, current smoking, anabolic steroid use, recent upper-respiratory infections or allergies, and reported snoring. F1 summarizes participant allocation, diagnostic protocols, and the types of voice recordings used across datasets (Fig 1).

**Speech acquisition**

To bring the algorithm closer to real-world conditions, the voice acoustic dataset was created using WA messages recorded and sent by the volunteers via their own cell phones in Brazilian Portuguese. It is worth mentioning that smartphone models were not controlled, considering the goal of naturalistic measurement. Participants used their own devices in their usual environments, which inherently introduced variability in microphone type, recording quality, and background

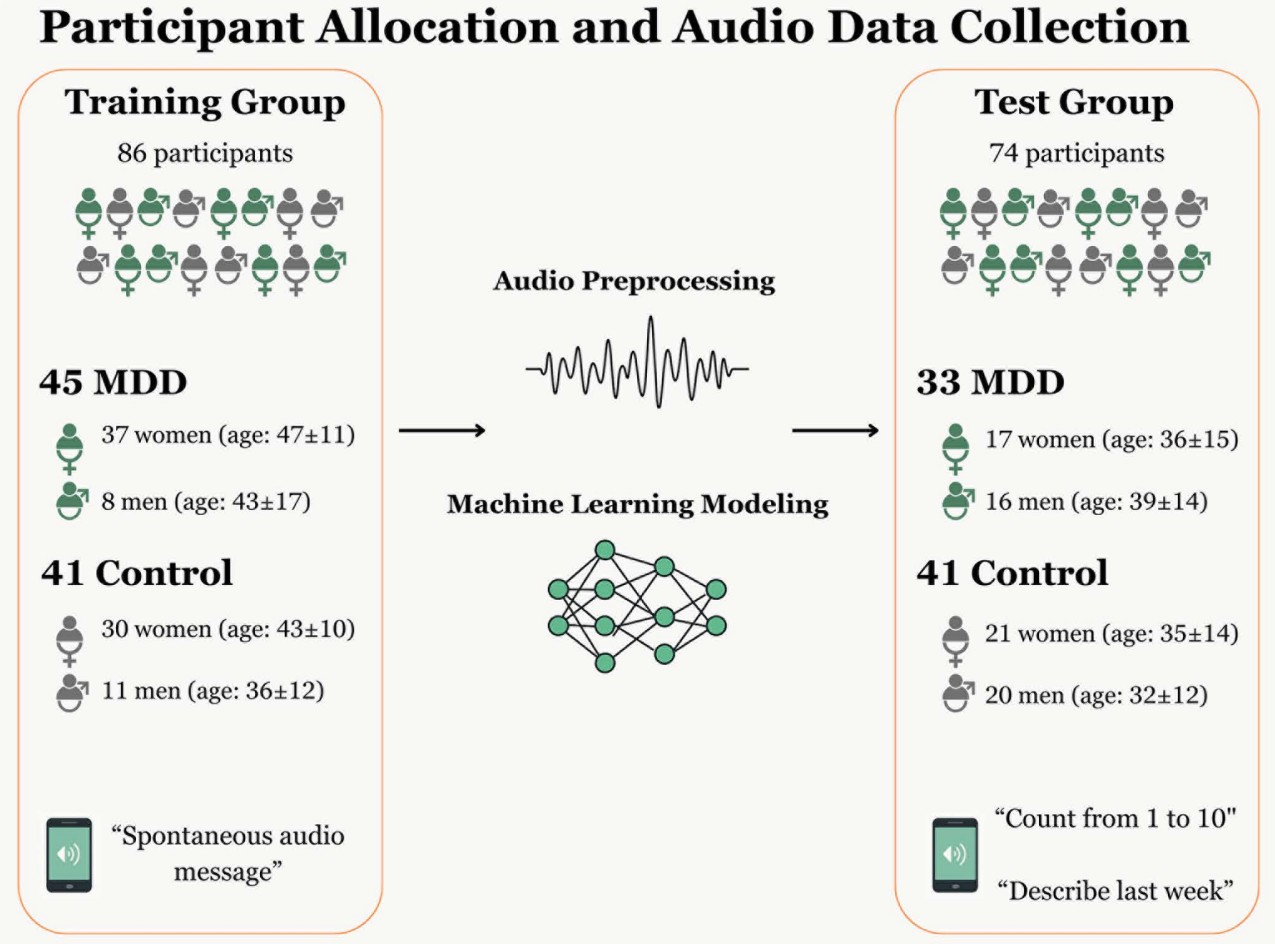

**Fig 1. Training and test group composition, and voice task types.** Participants were divided into a training group with spontaneous WhatsApp™ messages and a test group with structured tasks.

noise. When available, the general device type (e.g., Android vs. iOS) was documented, but no restrictions or adjustments were imposed to standardize hardware, in order to preserve ecological validity. To mitigate the impact of these uncontrolled factors, all audio underwent the same preprocessing pipeline described below, including silence removal and noise filtering.

A description of the WA message characteristics for each dataset is provided below:

i) In the training dataset, the speech data consisted of spontaneous WAs sent by patients to their doctors or secretaries when they were known to be symptomatic, and spontaneous WAs sent as part of routine activities, selected by the healthy volunteers themselves;

ii) In the test dataset, the speech data consisted of WAs following two protocols: a) volunteers recorded audio messages counting from 1 to 10 in a one-second rhythm, and b) audio messages describing how their past week had been (no minimum and maximum duration). Table 2 presents details about the number of audios in each task.

## Audio signal preprocessing

After.OGG files were converted to.WAV format with a 44.1kHz sampling rate, the audio signals included speech, silence and background ambient noise. In this study, we use the term sample to refer to the complete audio file obtained from each participant (e.g., a single WhatsApp message), and the term frame to refer to a short, fixed-duration segment of that sample generated during the feature extraction process. This distinction is maintained throughout the manuscript to ensure consistency and methodological clarity. The audio data from the training dataset was used to develop the ML models as internal validation, while the full audio files from the test dataset were utilized to evaluate the ML models' learning as external validation. Silence removal was performed using the Python library "*pydub*" (version 0.25.1) [20]. The *pydub* library is an audio processing tool that performs audio segmentation based on silence. In this study, 100 milliseconds were considered the minimum length for a silent passage, and the threshold for detecting silence was set at -40 decibels, with the Audacity manual [21] used as a reference. Furthermore, after processing, the audio files in the training sample set were limited to a maximum duration of 1 minute to ensure that longer audio files did not have greater weight. As shown in Table 3, the total duration of the original audio in the training set was 3673 seconds (146916 frames). After processing, it was reduced to 2509 seconds (96358 frames). For the test set, the original total duration was 4581 seconds (183248 frames), which was reduced to 3244 seconds (129781 frames).

## Machine learning models

A total of 68 acoustic features were extracted using the Python library PyAudioAnalysis [22]. Table 4 provides a brief description of these acoustic characteristics. Mel frequency cepstral coefficients (MFCCs) [22], widely used in automatic speech recognition [23] and depression detection [24–26], were utilized. Furthermore, the chroma feature, which considers spectral information, timbre, melody, rhythmic patterns and intonation, was also employed for speech recognition [27]. Table 4 summarizes the main extracted features.

**Table 2. Task audio sample information.**

| Task | Men Control | Women Control | Men Depression | Women Depression | Total |
|---|---|---|---|---|---|
| 1–10 | 20 | 21 | 15 | 16 | 72 |
| Last week | 20 | 21 | 16 | 15 | 72 |
| Total | 40 | 42 | 31 | 31 | 144 |

**Table 3. Information from the audio samples of the scaled and unscaled groups during the pre-processing stage.**

| Group | Original Audios | Post-Treatment Audios |
|---|---|---|
| Test | | |
| Number of Samples | 144 | 144 |
| Average Talking Time, standard deviation (s) | 32 (32) | 22 (28) |
| Total speaking time(s) | 4581 | 3244 |
| Frames Audios | 183248 | 129781 |
| Training | | |
| Number of Samples | 86 | 86 |
| Average Talking Time, standard deviation (s) | 43 (47) | 28 (19) |
| Total speaking time(s) | 3673 | 2509 |
| Frames Audios | 146916 | 96358 |

**Table 4. Acoustic features extracted by the PyAudioAnalysis library [4].**

| Index | Name | Description |
|---|---|---|
| 1 | Zero Crossing Rate | The rate of sign-changes of the signal during the duration of a particular frame. |
| 2 | Energy | The sum of squares of the signal values, normalised by the length. |
| 3 | Entropy of Energy | The entropy of sub-frames normalized energies. It can be interpreted as a measure of abrupt changes. |
| 4 | Spectral Centroid | The centre of gravity of the spectrum. |
| 5 | Spectral Spread | The second of gravity of the spectrum. |
| 6 | Spectral Entropy | Entropy of the normalized spectral energies for a set of sub-frames |
| 7 | Spectral Flux | The squared difference between the normalised magnitudes of the spectra of the two successive frames. |
| 8 | Spectral Rolloff | The frequency below which 90% of the magnitude distribution of the spectrum is concentrated. |
| 9-21 | MFCCs | Mel Frequency Cepstral Coefficients form a cepstral representation where the frequency bands are not linear but distributed according to the mel-scale. |
| 22-33 | Chroma Vector | A 12-element representation of the spectral energy where the bins represent the 12 equal-tempered pitch classes of western-type music (semitone spacing). |
| 34 | Chroma Deviation | The standard deviation of the 12 chroma coefficients. |

When deriving each of the characteristics to the first order, a total of 68 features are obtained. Adapted from Giannakopoulos T. (2015). pyAudioAnalysis: An Open-Source Python Library for Audio Signal Analysis. PLoS One, 10(12), e0144610. https://doi.org/10.1371/journal.pone.0144610.

 This study explored all these acoustic characteristics with the aim of determining whether, when combined, they could serve as biomarkers of depression, enabling the distinction between individuals in the control group and those with depression profile. Each audio sample in.WAV format was segmented into frames of 50ms with 25ms steps for the extraction of acoustic features.

 Seven machine learning models were employed: Adaboost (ADA), Decision Tree (DT), k-Nearest Neighbors (kNN), Linear Discriminant Analysis (LDA), Logistic Regression (LR) and Random Forest (RF), available on scikit-learn. Additionally, an Artificial Neural Network (ANN) was developed. The proposed ANN consisted of four layers: two fully

connected dense layers with ReLU activation function, and a final dense layer with a single neuron utilizing sigmoid activation function.

For the training and evaluation of the model, we employed a pipeline that integrates preprocessing techniques, class balancing and hyperparameter selection. First, the dataset was balanced using the NearMiss under sampling technique, which reduces the predominance of majority classes by selecting representative examples close to decision boundaries. After, normalization was applied using the StandardScaler, ensuring that the variables were scaled to have a mean of zero and a unit standard deviation.

The models were optimized using a Bayesian hyperparameter search implemented by the BayesSearchCV algorithm. To assess the model's performance impartially and avoid biases associated with hyperparameter tuning, we employed a nested cross-validation scheme. The validation process consists of two levels: internal and external. At the internal level, hyperparameter tuning was performed on five different training subsets using a stratified validation split. At the external level, the performance of the adjusted model was evaluated on five independent test sets, ensuring a robust estimate of generalization performance. Parallelly, in order to better understand the findings, a brief analysis was made to identify the 10 most important acoustic characteristics. From the trained models, the test set audios related to the tasks were used to evaluate the classification of the models. The detailed machine learning design is shown in Fig 2.

Model hyperparameters were optimized using a nested cross-validation (nested-CV) approach, which allows unbiased estimation of generalization performance by separating the processes of parameter tuning and performance evaluation (Vabalas et al., 2019).

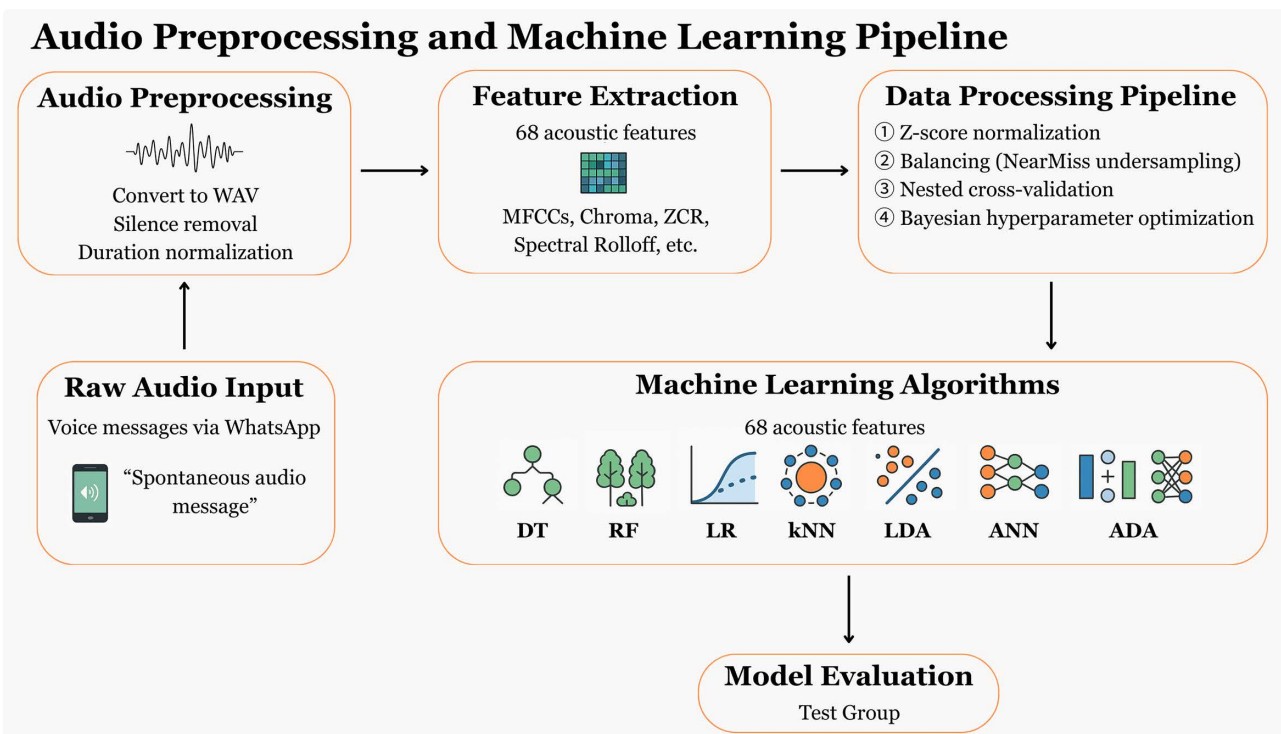

**Fig 2. Overview of preprocessing steps, feature extraction, and modeling process.** WhatsApp™ audios were preprocessed, acoustic features extracted, and models trained with a standardized pipeline. Seven ML algorithms were evaluated on an independent test set.

This study adopted the following statistical performance indicators: recall, specificity, positive predictive value (PPV), negative predictive value (NPV), accuracy, F1-score, and the area under the receiver operating characteristic (ROC) curve (AUC).

Performance metrics were computed at two analytical levels. During cross-validation (training phase), performance was evaluated at the frame level, for which Specificity and Negative Predictive Value (NPV) are not directly applicable due to the multi-frame structure of the data. In contrast, the external test phase employed audio-level aggregation, allowing calculation of all confusion matrix–based metrics, including Specificity and NPV. This distinction is clarified in the corresponding table legends.

Additionally, for the best-performing models, we calculated 95% confidence intervals (CIs) for accuracy, AUC, sensitivity, and specificity. All tests were two-tailed with a significance threshold of $p < 0.05$. Detailed statistical outputs, including all CIs, are available in S1 Text.

## Results

### Machine learning results

In this section, the classification performance of the machine learning models in detecting depression in audio samples from the test dataset is analyzed.

All classifiers were validated using the nested cross-validation strategy to optimize each model's hyperparameters. A standardized pipeline was applied across all machine learning estimators, incorporating the NearMiss method to balance the frame set, followed by standardization with the StandardScaler. Table 5 presents the best hyperparameters of each model and Table 6 presents the learning statistics of the models during the frame classification training process. Subsequently, an analysis using the Python library LIME was conducted to identify the 10 acoustic features that most influence the models' decisions as presented in Table 7.

The trained models were used to classify which WA in the test group belonged to participants with depression or healthy volunteers. Table 8 presents the classification of the models using only acoustic features (as described in Table 4), focusing on the algorithms with the highest AUC for each sex and speech protocol. The results demonstrated that, for the WA "how their past week had been", the LDA model in women achieved the highest AUC (91.9). In the same task, the LDA and LR models had the best result in men (AUC = 75). Furthermore, for the WA task of counting from 1 to 10, the RF model had the best AUC performance in women (82.74), while the ADA, LDA and RF models performed the best in men (AUC = 78.33).

Fig 3 and Fig 4 aim to illustrate the performance of different classification models in identifying individuals with depression in two tasks: "Last Week" (how their last week went) and "1to10" (count from 1 to 10). Each graph contains two groups on the x-axis: Healthy (individuals without depression) and Depressed (individuals diagnosed with depression). The y-axis represents the proportion of positive frames for depression, ranging from 0 (no frames associated with depression) to 1 (100% of the frames positive for depression). The graphs include key elements: black circles represent the proportion of positive frames for depression classified by the models for each individual, and the dashed red line indicating the cutoff line that separates the groups as optimally as possible for each model. In general, the models can capture the difference between the groups, Healthy and Depressed, with varying performance depending on gender, male or female, and the task.

By applying the Bootstrap method with 1000 iterations for each task and stratifying by sex, we obtained estimates with 95% confidence intervals (CI). In the "describe how their past week was" task for women, all models achieved mean AUC values above 0.85, well above the 0.50 chance level. The best-performing model was kNN (AUC = 0.945; CI: 0.869–0.996). In contrast, in the same task for men, AUCs ranged from 0.661 (ANN, CI: 0.475–0.836) to 0.758 (LR, CI: 0.575–0.909). For the "counting from 1 to 10" task in women, AUCs ranged between 0.715 (ANN, CI: 0.535–0.870) and 0.875 (kNN, CI: 0.743–0.970), with RF also performing strongly (AUC = 0.863; CI: 0.717–0.983). Finally, in the same task for

**Table 5. Optimization of model hyperparameters.**

| Model | Space Parameters | Best Parameters |
|---|---|---|
| **DT** | **criterion:** gini, entropy | **criterion:** entropy |
| | **max_depth:** (5, 10) | **max_depth:** 9 |
| | **min_samples_splits:** (5, 30) | **min_samples_splits:** 25 |
| | **min_sample_leaf:** (5,30) | **min_sample_leaf:** 18 |
| | **max_features:** sqrt, log2 | **max_features:** log2 |
| | **splitter:** best, random | **splitter:** best |
| **Adaboost** | **algorithm:** SAMME, SAMME.R | **algorithm:** SAMME.R |
| | **learning_rate:** (0.01, 1.0) | **learning_rate:** 0.69186037 |
| | **n_estimators:** (10, 200) | **n_estimators:** 104 |
| **LR** | **C:** (0.001, 0.01, 0.1, 1, 10) | **C:** 10 |
| | **max_iter:** (100, 300) | **max_iter:** 731 |
| | **solver:** newton-cg, lbfgs, liblinear | **solver:** newton-cg |
| | **penalty:** l1, l2 | **penalty:** l2 |
| **kNN** | **n_neighbors:** (1, 50) | **n_neighbors:** 22 |
| | **weights:** uniform, distance | **weights:** distance |
| | **metric:** euclidean, manhattan | **metric:** manhattan |
| **RF** | **n_estimators:** (10,101) | **n_estimators:** 29 |
| | **max_depth:** (3, 6) | **max_depth:** 6 |
| | **min_samples_split:** (32, 129) | **min_samples_split:** 69 |
| | **min_samples_leaf:** (32, 129) | **min_samples_leaf:** 72 |
| | **bootstrap:** True or False | **bootstrap:** False |
| | **criterion:** gini, entropy | **criterion:** entropy |
| **ANN** | **epochs:** 15 | **epochs:** 15 |
| | **batch_size:** (8, 16, 32) | **batch_size:** 16 |
| | **optimizer:** adam | **optimizer:** adam |
| | **learning_rate:** (1e-4, 1e-3, 1e-2, 1e-1) | **learning_rate:** 1e-3 |
| | **dense_1_units:** (16, 32, 64, 128) | **dense_1_units:** 64 |
| | **dense_2_units:** (16, 32, 64, 128) | **dense_2_units:** 128 |
| **LDA** | **shrinkage:** (1e-05, 1e-04, 1e-03, 1e-2, 1e-1,1) | **shrinkage:** 1e-05 |
| | **solver:** lsqr, eigen, svd | **solver:** eigen |
| | **tol:** (1e-05, 1e-01) | **tol:** 0.0138147905 |

**Table 6. Statistics of the training process of the models during cross-validation (mean±2*SD).**

| Models | Recall | PPV | Accuracy | F1-Score | AUC |
|---|---|---|---|---|---|
| kNN | 75.63±0.27 | 73.41±0.13 | 66.87±0.12 | 74.50±0.11 | 69.27±0.26 |
| LDA | 50.96±0.63 | 50.96±0.63 | 55.15±0.31 | 59.26±0.46 | 58.94±0.47 |
| LR | 50.78±0.58 | 70.76±0.18 | 55.07±0.29 | 59.13±0.43 | 58.71±0.48 |
| DT | 58.02±3.38 | 67.94±0.59 | 55.58±0.87 | 62.52±1.83 | 56.99±0.20 |
| Adaboost | 51.03±2.73 | 69.41±1.10 | 54.21±0.30 | 58.75±1.35 | 54.21±0.30 |
| RF | 52.83±0.50 | 70.19±0.27 | 55.45±0.13 | 60.28±0.26 | 58.06±0.25 |
| ANN | 58.90±0.98 | 75.56±0.75 | 61.49±0.25 | 66.19±0.40 | 65.78±0.44 |

Performance metrics were computed at the frame level, where Specificity and Negative Predictive Value (NPV) are not directly applicable. Legend: Artificial Neural Network (ANN), Decision Tree (DT), k-Nearest Neighbors (kNN), Linear Discriminant Analysis (LDA), Logistic Regression (LR), Random Forest (RF), Positive Predictive Value (PPV), and Area Under the Curve (AUC).

**Table 7.** The table presents the top 10 acoustic features with the highest importance in the decision-making process of the machine learning models.

| Feature Ranking | Models | | | | | | |
|---|---|---|---|---|---|---|---|
| | Adaboost | ANN | DT | kNN | LDA | LR | RF |
| 1 | MFCC 2 | MFCC 2 | Delta MFCC 1 | MFCC 10 | MFCC 2 | MFCC 2 | MFCC 2 |
| 2 | Delta MFCC 1 | Spectral Rolloff | ZCR | Chroma 10 | Spectral Rolloff | Spectral Rolloff | Delta MFCC 1 |
| 3 | Chroma 10 | Chroma 7 | MFCC 6 | MFCC 2 | ZCR | ZCR | ZCR |
| 4 | Spectral Flux | MFCC 10 | Spectral Rolloff | MFCC 4 | Chroma 7 | Chroma 7 | Spectral Flux |
| 5 | MFCC 10 | Chroma 7 | Chroma 7 | MFCC 6 | MFCC 1 | MFCC 1 | Spectral Rolloff |
| 6 | Chroma 7 | Delta MFCC 1 | Energy Entropy | Chroma 7 | MFCC 10 | Spectral Flux | Chroma 10 |
| 7 | MFCC 1 | Spectral Flux | Energy | Delta Chroma 7 | Spectral Flux | MFCC 10 | Chroma 7 |
| 8 | Delta MFCC 3 | Spectral Entropy | MFCC 8 | MFCC 7 | Spectral Entropy | Chroma 11 | Spectral Entropy |
| 9 | ZCR | ZCR | MFCC 2 | Chroma 11 | Chroma 11 | Spectral Entropy | Spectral Spread |
| 10 | Spectral Rolloff | Chroma 4 | Chroma std | Chroma std | Chroma 1 | Delta Energy | Energy Entropy |

men, performance was lower, with AUC values ranging from 0.596 (DT, CI: 0.401–0.803) to 0.796 (RF, CI: 0.632–0.942). Table 9 summarizes the AUC-ROC results along with False Positive Rate (FPR) and False Negative Rate (FNR), while detailed results for additional metrics (recall, specificity, PPV, NPV, accuracy, and F1-score) are provided in S3 Text.

For the best-performing models in each task and gender subgroup, 95% confidence intervals (CIs) were calculated for accuracy, AUC, sensitivity, and specificity. Statistical comparisons between models and between male and female subgroups were performed using DeLong's test for AUC differences and two-proportion z-tests for accuracy differences. Full statistical outputs, including all CIs, are presented in S3 Text.

Model performance varied slightly between male and female participants, with higher accuracy and AUC values observed among women. This difference likely reflects the gender imbalance in the training dataset, where the smaller number of male participants increased variance and widened confidence intervals. To preserve ecological validity, we did not apply oversampling or reweighting procedures, instead reporting gender-stratified results and discussing this limitation explicitly below.

Additionally, false positive rates (FPR) and false negative rates (FNR) were computed from the confusion matrices for each task and gender. Some recurrent misclassifications included false positives in recordings with high background noise or overlapping speech, and false negatives in cases where depressed participants produced speech with a neutral or monotone prosody. Detailed confusion matrices (absolute and normalized), along with FPR/FNR values, are available in S1 Text.

To assess the statistical robustness of the classification models, we applied the Mann–Whitney test available in the Python scipy.stats library to compare the classification distributions between groups, reporting p-values, effect sizes (Cohen's d), and mean percentage differences (S4 Text).

It is worth noting that training metrics were computed at the frame level—reflecting the model's ability to classify short audio segments as depressive or non-depressive—whereas test performance was derived from aggregated audio-level probabilities (i.e., the proportion of frames classified as depressive per recording). This methodological distinction can lead to slightly higher apparent test performance, as aggregation averages out frame-level noise and misclassifications.

## Discussion

The objective of this study was to evaluate the performance of machine learning (ML) models in classifying the depressive profile of Brazilian individuals using vocal signals extracted from WhatsApp™ audio messages. Participants either described how their previous week went or counted from 1 to 10. The findings suggest that the acoustic features of speech extracted from these audios have potential as biomarkers for detecting depressive profiles through ML.

**Table 8. Statistical results of the best-performing models using 68 acoustic features extracted from test audio samples in two different tasks. Metrics were computed at the audio level, allowing inclusion of all confusion matrix–derived parameters.**

Task "describe how their past week was" audios (female)

| Models | Recall | Specificity | PPV | NPV | Accuracy | F1-Score | AUC |
|---|---|---|---|---|---|---|---|
| ADA | 86.67 | 85.71 | 81.25 | 90 | 86.11 | 83.87 | 86.19 |
| ANN | 73.33 | 95.24 | 91.67 | 83.33 | 86.11 | 81.48 | 84.28 |
| DT | 80 | 85.71 | 80 | 85.71 | 83.33 | 80 | 82.86 |
| kNN | 86.67 | 85.71 | 81.25 | 90 | 86.11 | 83.87 | 86.19 |
| LDA | 93.33 | 90.48 | 87.5 | 95 | 91.67 | 90.32 | 91.9 |
| LR | 86.67 | 90.48 | 86.67 | 90.48 | 88.89 | 86.67 | 88.57 |
| RF | 80 | 95.24 | 92.31 | 86.96 | 88.89 | 85.71 | 87.62 |

Task "describe how their past week was" audios (male)

| Models | Recall | Specificity | PPV | NPV | Accuracy | F1-Score | AUC |
|---|---|---|---|---|---|---|---|
| ADA | 62.5 | 80 | 71.43 | 72.73 | 72.22 | 66.67 | 71.25 |
| ANN | 81.25 | 55 | 59.09 | 78.57 | 66.67 | 68.42 | 68.12 |
| DT | 68.75 | 75 | 68.75 | 75 | 72.22 | 68.75 | 71.87 |
| kNN | 56.25 | 65 | 56.25 | 65 | 61.11 | 56.25 | 60.62 |
| LDA | 75 | 75 | 70.59 | 78.95 | 75 | 72.73 | 75 |
| LR | 75 | 75 | 70.59 | 78.95 | 75 | 72.73 | 75 |
| RF | 68.75 | 80 | 73.33 | 76.19 | 75 | 70.97 | 74.37 |

Task "counting from 1 to 10" audios (female)

| Models | Recall | Specificity | PPV | NPV | Accuracy | F1-Score | AUC |
|---|---|---|---|---|---|---|---|
| ADA | 93.75 | 52.38 | 60 | 91.67 | 70.27 | 73.17 | 73.06 |
| ANN | 62.5 | 80.95 | 71.43 | 73.91 | 72.97 | 66.67 | 71.73 |
| DT | 62.5 | 90.48 | 83.33 | 76 | 78.38 | 71.43 | 76.49 |
| kNN | 93.75 | 71.43 | 71.43 | 93.75 | 81.08 | 81.08 | 82.59 |
| LDA | 81.25 | 76.19 | 72.22 | 84.21 | 78.38 | 76.47 | 78.72 |
| LR | 81.25 | 76.19 | 72.22 | 84.21 | 78.38 | 76.47 | 78.72 |
| RF | 75 | 90.48 | 85.71 | 82.61 | 83.73 | 80 | 82.74 |

Task "counting from 1 to 10" audios (male)

| Models | Recall | Specificity | PPV | NPV | Accuracy | F1-Score | AUC |
|---|---|---|---|---|---|---|---|
| ADA | 86.67 | 70 | 68.42 | 87.5 | 77.14 | 76.47 | 78.33 |
| ANN | 66.67 | 65 | 58.82 | 72.22 | 65.71 | 62.5 | 65.83 |
| DT | 46.67 | 80 | 63.64 | 66.67 | 65.71 | 53.84 | 63.33 |
| kNN | 80 | 50 | 54.54 | 76.92 | 62.85 | 64.86 | 65 |
| LDA | 66.67 | 90 | 83.33 | 78.26 | 80 | 74.07 | 78.33 |
| LR | 60 | 95 | 90 | 76 | 80 | 72 | 77.5 |
| RF | 86.67 | 70 | 68.42 | 87.5 | 77.14 | 76.47 | 78.33 |

Legend: Adaboost (ADA), Artificial Neural Network (ANN), Decision Tree (DT), k-Nearest Neighbors (kNN), Linear Discriminant Analysis (LDA), Logistic Regression (LR), Random Forest (RF), Positive Predictive Value (PPV), Negative Predictive Value (NPV), and Area Under the Curve (AUC).

It is important to note that our models were intentionally designed for binary pre-screening purposes, aimed at differentiating individuals with depression from healthy controls. While this approach provides an essential proof of feasibility, we recognize that depression manifests along a continuum of severity and subtypes. Accordingly, future work will extend this framework toward ordinal and regression-based analyses capable of capturing symptom gradients and classifying scores on standardized scales, thereby bridging the gap between early screening and clinical phenotyping.

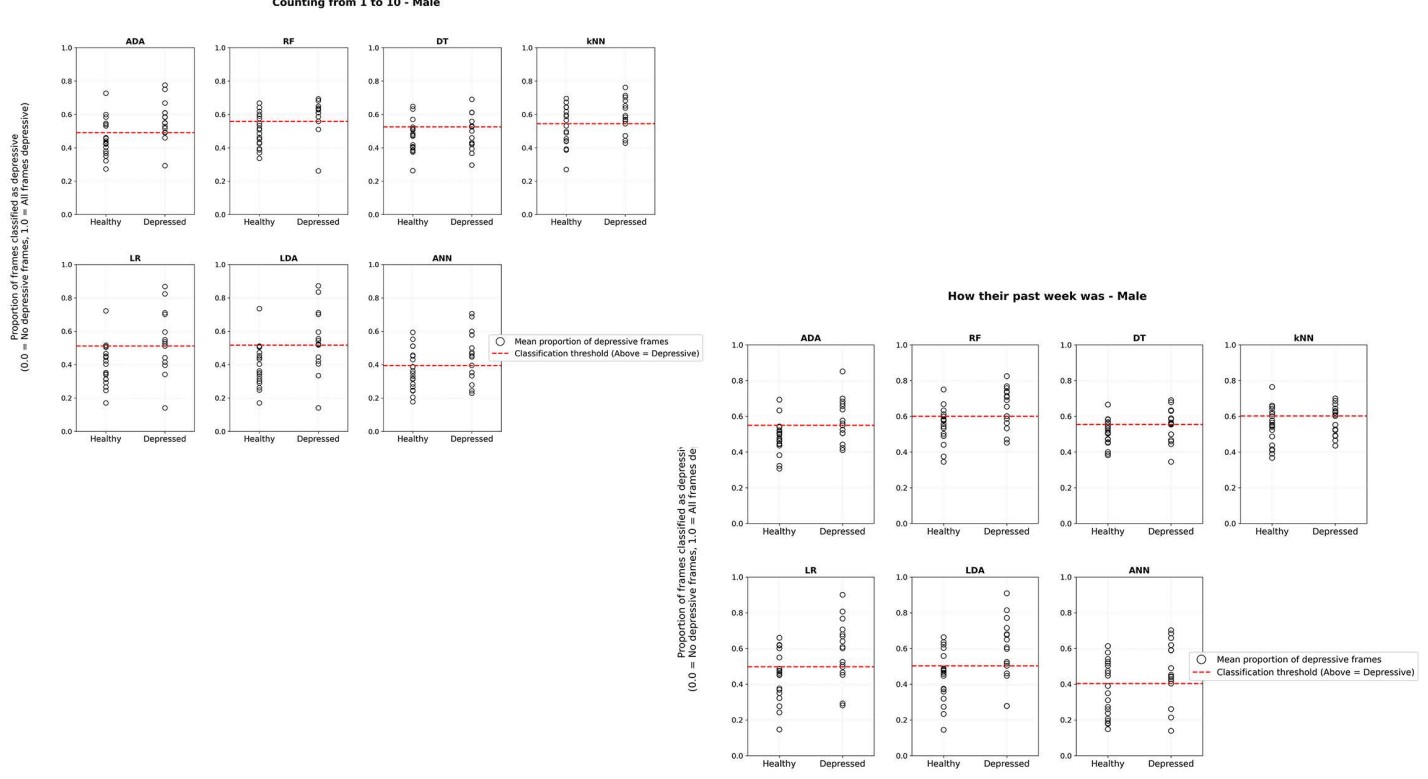

**Fig 3. Results of ML models applied to the test dataset using recordings of male voices.** The audio files are distributed according to the proportion of positive frames, ranging from the lowest to the highest value in each group (Healthy and Depressed).

## ML models performance

The study aims to internally and externally validate different ML models to ensure the robustness of the results. Internal validation was conducted using a set of audio recordings from healthy and depressed participants, collected by psychiatrists and medical residents. Interestingly, in 5 out of the 7 models (including the 9th decision tree (DT) and the 3rd k-nearest neighbor (kNN)), the most important feature was MFCC 2, which is consistent with the findings of Taguchi T et al., who also identified the relevance of this feature in studies with Japanese participants.

The prominence of MFCC2 across multiple classifiers may have a biological basis related to how depression affects speech production. MFCC2 represents low-frequency components of the spectral envelope, which are influenced by the configuration of the vocal tract and articulatory movements during phonation. These parameters can be modulated by respiratory-laryngeal coordination, subglottal pressure, and fine motor control of the articulators — all of which may be affected in depressive states due to psychomotor slowing, reduced prosodic variation, and changes in muscle tension. Such alterations can result in shifts in formant structure and spectral energy distribution that MFCC2 is sensitive to. Other high-ranking features in our models, such as spectral rolloff and zero-crossing rate, may capture related phenomena, including reductions in high-frequency energy and flatter pitch contours, both of which are consistent with previously reported acoustic profiles of depressed speech. By linking these features to known physiological and behavioral manifestations of depression, we provide a basis for their potential use as biomarkers in clinical applications. When examining feature stability across validation folds, MFCC 2 consistently appeared among the top-ranked features in all models, confirming its robustness across both training and external test datasets. Other features, such as spectral rolloff

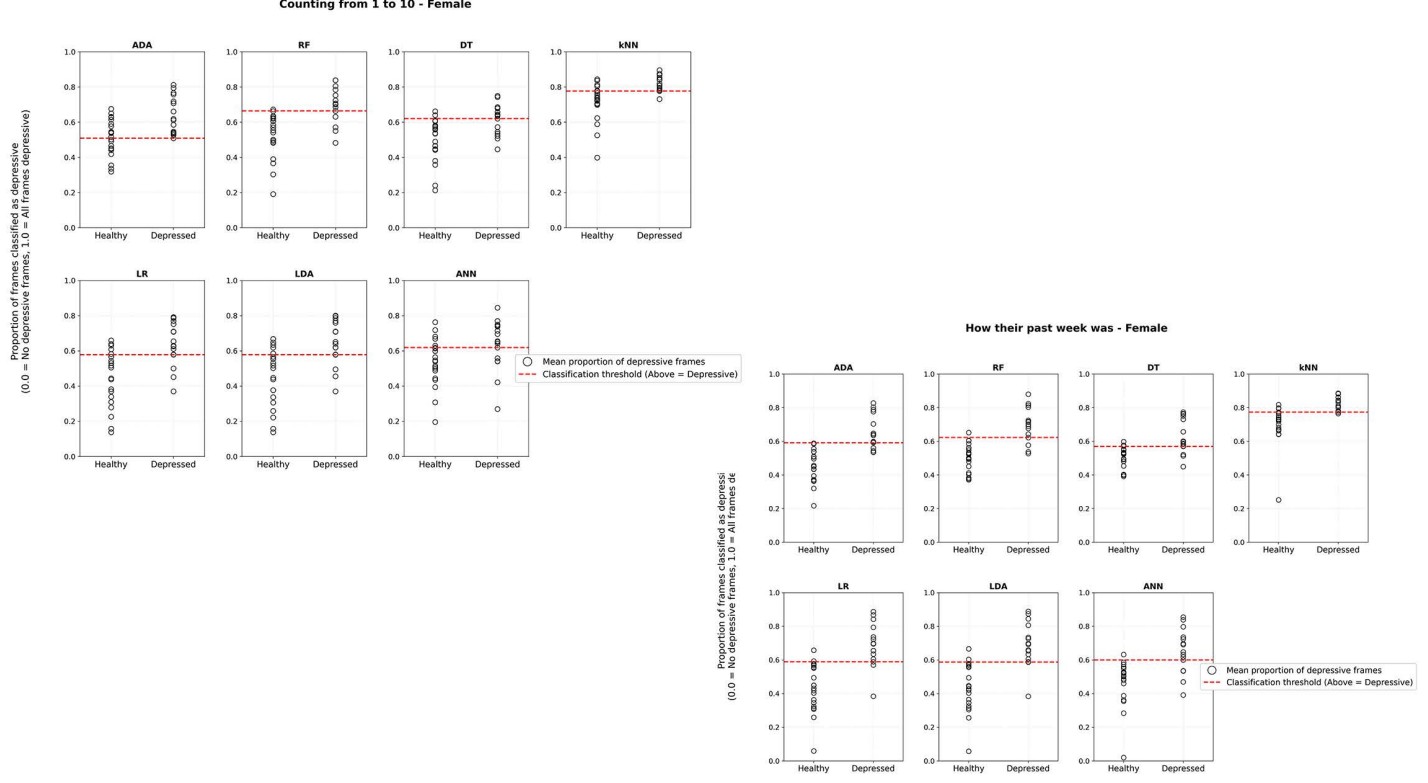

**Fig 4. Results of ML models applied to the test dataset using recordings of female voices.** The audio files are distributed according to the proportion of positive frames, ranging from the lowest to the highest value in each group (Healthy and Depressed).

and zero-crossing rate, were recurrent but not uniformly present across folds, which may reflect inter-individual variability, differences in recording conditions, and the intentional domain shift between datasets. This variability suggests that depression-related acoustic signatures are likely distributed across multiple complementary features, supporting the value of multi-feature approaches for robust classifying modeling.

For the external validation of the ML models, we applied the algorithms to an independent set of audio recordings from healthy and depressed individuals (MADRS: mean = 26; SD = 9). This sample was collected by psychiatrists and residents who were not involved in the training dataset. Among female participants, the lowest AUC was 0.71 in the task "describe how your past week was," which is significantly above chance level (AUC = 0.50). Moreover, the AdaBoost and kNN models achieved AUCs of up to 0.94. Among male participants, the Random Forest and AdaBoost models yielded the best performance in the "count from 1 to 10" task, with AUCs of approximately 0.80. These results may be associated with social factors, as traditional masculine norms often make it more difficult for men to express their feelings (Seidler ZE et al., 2016). As a result, the superior performance in the "count from 1 to 10" task may be explained by its more objective and less emotionally demanding nature.

An important methodological aspect of this study is the intentional domain shift between the training set, which comprised spontaneous WhatsApp™ messages, and the testing set, which used semi-structured speech tasks ("count from 1 to 10" and "describe how your past week was"). This choice was made to simulate real-world clinical scenarios, where training data may be derived from natural patient communications while assessment tasks can be more structured. While this design allowed us to evaluate the models' ability to generalize across different speech styles, it may also introduce

**Table 9. Statistical results applying the bootstrap method with 1000 iterations, mean and 95% confidence interval.**

**Task "describe how their past week was" audios (female)**

| Models | AUC | FPR | FNR |
| --- | --- | --- | --- |
| ADA | 0.945 (0.869-0.997) | 0.092 (0.000-0.318) | 0.106 (0.000-0.312) |
| ANN | 0.860 (0.711-0.981) | 0.078 (0.000-0.276) | 0.207 (0.000-0.467) |
| DT | 0.851 (0.694-0.977) | 0.089 (0.000-0.273) | 0.218 (0.000-0.462) |
| kNN | 0.944 (0.864-0.996) | 0.125 (0.000-0.333) | 0.057 (0.000-0.250) |
| LDA | 0.931 (0.819-1.000) | 0.077 (0.000-0.218) | 0.073 (0.000-0.231) |
| LR | 0.925 (0.812-1.000) | 0.081 (0.000-0.240) | 0.104 (0.000-0.294) |
| RF | 0.925 (0.815-0.997) | 0.059 (0.000-0.320) | 0.160 (0.000-0.385) |

**Task "describe how their past week was" audios (male)**

| Models | AUC | FPR | FNR |
| --- | --- | --- | --- |
| ADA | 0.745 (0.571-0.910) | 0.154 (0.000-0.409) | 0.288 (0.059-0.562) |
| ANN | 0.661 (0.475-0.836) | 0.253 (0.000-0.632) | 0.327 (0.000-0.750) |
| DT | 0.673 (0.466-0.855) | 0.180 (0.000-0.412) | 0.356 (0.071-0.688) |
| kNN | 0.599 (0.401-0.790) | 0.383 (0.000-0.857) | 0.291 (0.000-0.778) |
| LDA | 0.746 (0.574-0.898) | 0.171 (0.000-0.409) | 0.303 (0.049-0.667) |
| LR | 0.758 (0.575-0.909) | 0.168 (0.000-0.429) | 0.295 (0.053-0.643) |
| RF | 0.753 (0.571-0.909) | 0.135 (0.000-0.360) | 0.320 (0.067-0.600) |

**Task "counting from 1 to 10" audios (female)**

| Models | AUC | FPR | FNR |
| --- | --- | --- | --- |
| ADA | 0.786 (0.624-0.913) | 0.319 (0.000-0.619) | 0.139 (0.000-0.588) |
| ANN | 0.715 (0.535-0.870) | 0.207 (0.000-0.560) | 0.305 (0.000-0.625) |
| DT | 0.786 (0.623-0.918) | 0.161 (0.000-0.550) | 0.283 (0.000-0.562) |
| kNN | 0.875 (0.743-0.970) | 0.210 (0.000-0.400) | 0.071 (0.000-0.267) |
| LDA | 0.828 (0.673-0.944) | 0.174 (0.000-0.500) | 0.220 (0.000-0.563) |
| LR | 0.836 (0.684-0.952) | 0.158 (0.000-0.429) | 0.216 (0.000-0.538) |
| RF | 0.863 (0.717-0.983) | 0.039 (0.000-0.174) | 0.242 (0.000-0.500) |

**Task "counting from 1 to 10" audios (male)**

| Models | AUC | FPR | FNR |
| --- | --- | --- | --- |
| ADA | 0.787 (0.618-0.938) | 0.273 (0.045-0.500) | 0.120 (0.000-0.417) |
| ANN | 0.660 (0.470-0.829) | 0.253 (0.000-0.688) | 0.350 (0.000-0.769) |
| DT | 0.596 (0.401-0.803) | 0.262 (0.000-0.680) | 0.380 (0.000-0.750) |
| kNN | 0.654 (0.454-0.826) | 0.342 (0.000-0.762) | 0.271 (0.000-0.769) |
| LDA | 0.779 (0.587-0.932) | 0.080 (0.000-0.421) | 0.296 (0.050-0.556) |
| LR | 0.766 (0.587-0.931) | 0.097 (0.000-0.455) | 0.310 (0.000-0.571) |
| RF | 0.796 (0.632-0.942) | 0.233 (0.000-0.480) | 0.154 (0.000-0.400) |

Legend: Adaboost (ADA), Artificial Neural Network (ANN), Decision Tree (DT), k-Nearest Neighbors (kNN), Linear Discriminant Analysis (LDA), Logistic Regression (LR) and Random Forest (RF), False Positive Rate (FPR) and False Negative Rate (FNR), and Area Under the Curve (AUC).

bias, with part of the classification signal reflecting differences in speech structure rather than depressive status. We have therefore interpreted the results within this context and highlight that future work should include matched-speech validation and cross-task analyses to better isolate the effect of speech style on model performance.

Differences in diagnostic instruments, speech content, and demographic profiles between the training and test datasets likely contributed to observed variations in model performance. The training data involved spontaneous speech obtained under naturalistic conditions, whereas the test dataset comprised semi-structured speech tasks. This intentional heterogeneity was designed to assess domain generalization—reflecting the real-world variability of telehealth and clinical screening environments—but may also have introduced a domain-shift effect. Similarly, the use of distinct diagnostic instruments (DSM-5–based vs. MINI) and the slightly older average age in the test set, particularly among female participants, may have influenced classification accuracy. Future work will address these issues by harmonizing diagnostic protocols, age distributions, and speech content across datasets.

While our use of distinct diagnostic tools and speech tasks allowed us to test the robustness of model performance under domain-shifted conditions, this heterogeneity also introduces mild inconsistencies across datasets. Future work will focus on harmonizing diagnostic protocols, speech content, and demographic distributions to enhance dataset compatibility and improve the reproducibility of cross-corpus validations.

Social and cultural factors—particularly traditional masculine norms—may influence vocal expressivity and emotional communication, contributing to flatter prosody and reduced pitch variability among men. These tendencies can attenuate the acoustic cues associated with depressive affect, helping explain the lower classification accuracy observed in male participants. Previous studies have consistently shown that conformity to masculine norms is linked to lower emotional expression and reduced help-seeking behaviors (Addis & Mahalik, 2003; Seidler et al., 2016; Rice et al., 2023). Such expressive attenuation may obscure acoustic cues that typically differentiate depressive from neutral speech, resulting in subtler detectable patterns in men. These findings align with evidence from affective neuroscience indicating that emotion expression is shaped by both biological and cultural constraints. Future studies with larger and gender-balanced samples will be necessary to disentangle these sociocultural and physiological influences on voice-based biomarkers of depression.

### Innovation & implications

The human voice holds significant potential as a biomarker given its simple, non-invasive, and easily accessible nature. The analysis of its acoustic parameters through ML algorithms offers promising avenues for the early detection of depression and the assessment of therapeutic responses. In contemporary Brazilian society, audio communication, particularly via mobile devices and the WhatsApp™ application, is a ubiquitous aspect of daily life.

The psychiatry ability to diagnose and treat mental illnesses has been hindered by the absence of objective clinical tests, such as those routinely used in other fields of medicine. Diagnostic scales have limitations, including a lack of objectivity, inconsistent or infrequent results, and low specificity in identifying depression subtypes. Consequently, the diagnosis of mental disorders relies heavily on the clinician's experience and extensive training. A common instrument used for Depression screening, such as PHQ-9, typically presents an average sensitivity of 85% and specificity of 85%, considering the cut-off value of ≥10 (14), similar to what we found here, using our ML models. In this context, the use of a practical and low-cost biomarker, such as WhatsApp™ audio messages, could provide significant benefits for the screening of depressive conditions. Depression is a common and heterogeneous disorder with various symptomatic subtypes. The application of ML techniques in voice processing can help identify acoustic parameters classifying depression with melancholic and anxious features, while also improving the understanding of the distinct pathophysiology and clinical presentations of this disease. This analysis could aid in optimizing pharmacological therapies for these conditions. Additionally, this technique holds potential for detecting acoustic parameters classifying both disease severity and treatment response.

### Limitations

This study has several limitations that may affect the generalizability of its findings. The algorithm was trained on data from native Brazilian Portuguese speakers, primarily from a few cities, limiting its applicability across Brazil's diverse

linguistic and sociodemographic profiles (nevertheless, the data was captured from different Brazilian states, in order to better represent Brazilian linguistic diversity). Imbalances in the sample, including a higher proportion of female participants, resulted in better model accuracy for women's voices, indicating the need for additional data, particularly for male voices and underrepresented groups such as transgender and LGBTQ+ individuals. Rather than artificially balancing the dataset or applying re-weighting, we deliberately presented and interpreted results separately for men and women, as our aim in this initial study was to evaluate feasibility under naturalistic recruitment conditions, preserving ecological validity. Furthermore, voice features may be influenced by confounding factors such as hormonal cycles, mood variations, obesity, medication use, and psychiatric comorbidities, even after excluding major medical and neurological conditions. Although the present design introduces variability in diagnostic instruments and recording contexts, such heterogeneity also represents a methodological strength. It reflects real-world conditions under which voice-based screening tools are likely to operate, thereby enhancing ecological validity and potential generalizability. This intentional trade-off between control and realism should be considered when interpreting the results and planning future studies aimed at cross-corpus harmonization. In practical applications, it is unrealistic to expect all recordings to be collected in acoustic booths or with standardized high-quality microphones; instead, models must be robust to the diversity of devices, environments, and contexts encountered in everyday clinical and community scenarios. This ecological approach enhances the translational potential of our findings and provides a more accurate picture of how such a tool would function in actual deployment.

In addition, we conducted exploratory analyses stratified by gender to further examine the observed differences in model performance. These analyses included ROC curves for each task and gender combination, as well as absolute and normalized confusion matrices, which are provided in S2 Text and S1 Text, respectively. Although these results illustrate differences between male and female participants—particularly in the tasks "Counting from 1 to 10" and "Describing how their past week was"—the male subgroup size was limited, and thus these comparisons are underpowered. For this reason, the pooled results remain the primary focus of our conclusions, and the gender-stratified findings should be interpreted as preliminary and hypothesis-generating.

Gender imbalance represents a key limitation of the present study, particularly affecting model variance and generalization in male participants. The observed performance difference between genders highlights the need for larger, gender-balanced datasets in future research to enable fair and robust cross-gender validation. Moreover, methodological differences between frame-level training metrics and audio-level test aggregation should be considered when interpreting the comparative performance values.

Additionally, the diagnostic procedures differed between datasets: DSM-5-based clinical evaluations for the training set and the MINI structured interview for the testing set. While both methods are validated and widely used, this lack of harmonization in diagnostic criteria may have contributed to variability in model performance. Future prospective studies should aim to apply a single standardized diagnostic protocol across datasets to ensure full comparability.

Lastly, Performance differences between datasets may partly result from non-uniform diagnostic instruments (DSM-5 vs. MINI), varied speech tasks, and demographic discrepancies—especially the higher mean age in the test sample. These variations reflect the real-world heterogeneity of data sources but also highlight the importance of harmonized and demographically balanced datasets in future research. Also, the datasets used in this study differed slightly in diagnostic procedures and recording tasks, which, although intentional for assessing ecological validity, may limit strict cross-corpus comparability. Future data collections will incorporate harmonized assessment protocols and standardized speech prompts to minimize such differences and strengthen external validity.

## Conclusions

This study demonstrates the potential of machine learning models to screen/identify individuals with varying depressive profiles using a simple, accessible, and real-world classifiers: WhatsApp™ voice messages. By supporting the idea that voice can serve as a biomarker for depressive profiles, the research highlights a unique advantage—this biomarker likely

reflects the consequences of depression, rather than being entangled in the complex psychosocial dynamics of its pathophysiology. Remarkably, the accuracy achieved is comparable to that of established tools like the PHQ-9 [15], underscoring its promise for practical applications. While limitations exist regarding sample diversity and environmental control, this study provides robust evidence to support further exploration of the relationship between voice data and depression, paving the way for innovative clinical and research applications.

## Supporting information

**S1 Text. Classification by AUC and statistical significance for all models and tasks.** This file presents detailed statistical comparisons of classification models by AUC values, p-values, Cohen's d, and percentage differences for each speech task ("How their past week was" and "Counting from 1 to 10"), stratified by gender (female and male). The file also includes interpretation legends for significance and effect sizes.
(DOCX)

**S2 Text. Model performance metrics and confidence intervals.** Contains comprehensive performance tables for all tested models (AdaBoost, ANN, Decision Tree, kNN, LDA, Logistic Regression, Random Forest) including recall, specificity, PPV, NPV, accuracy, F1-score, AUC, and error rates with confidence intervals. The results are shown separately for female and male participants across both speech tasks.
(DOCX)

**S3 Text. Receiver Operating Characteristic (ROC) curves by gender and task.** Includes the ROC curve plots for each model and speech task ("Counting from 1 to 10" and "How their past week was"), divided by gender (female and male). These visualizations illustrate model discrimination performance across the different classification settings.
(DOCX)

**S4 Text. Confusion matrices and model accuracy visualization.** Presents absolute and normalized confusion matrices for each classification model, gender, and speech task. These matrices illustrate prediction distributions and support interpretation of model precision and recall performance.
(DOCX)

## Acknowledgments

The authors would like to express their gratitude to Fernando Janson for his support with the English language review.

## Author contributions

**Conceptualization:** Victor H. O. Otani, Ricardo R. Uchida.

**Data curation:** Victor H. O. Otani, Felipe O. Aguiar, Thiago P. Justino, Hudson S. Buck, Luiza B. Grilo, Matheus F. Figueiredo, Pedro M. Uchida, Daniel A. C. Vasques, Thaís Z. S. Otani, Lucas M. Marques, Ricardo R. Uchida.

**Formal analysis:** Victor H. O. Otani, Felipe O. Aguiar, Hudson S. Buck, Daniel A. C. Vasques, Thaís Z. S. Otani, João Ricardo N. Vissoci, Lucas M. Marques, Ricardo R. Uchida.

**Funding acquisition:** Victor H. O. Otani, Daniel A. C. Vasques, Thaís Z. S. Otani, Ricardo R. Uchida.

**Investigation:** Victor H. O. Otani, Felipe O. Aguiar, Thiago P. Justino, Hudson S. Buck, Luiza B. Grilo, Matheus F. Figueiredo, Pedro M. Uchida, Daniel A. C. Vasques, Thaís Z. S. Otani, Lucas M. Marques, Ricardo R. Uchida.

**Methodology:** Victor H. O. Otani, Felipe O. Aguiar, Thiago P. Justino, Hudson S. Buck, Luiza B. Grilo, Matheus F. Figueiredo, Pedro M. Uchida, João Ricardo N. Vissoci, Lucas M. Marques, Ricardo R. Uchida.

**Project administration:** Victor H. O. Otani, Felipe O. Aguiar, Lucas M. Marques, Ricardo R. Uchida.

**Resources:** Victor H. O. Otani, Felipe O. Aguiar, Thiago P. Justino, Hudson S. Buck, Luiza B. Grilo, Matheus F. Figueiredo, Pedro M. Uchida, Daniel A. C. Vasques, Thaís Z. S. Otani, Lucas M. Marques, Ricardo R. Uchida.

**Software:** Felipe O. Aguiar, Hudson S. Buck, João Ricardo N. Vissoci.

**Supervision:** Victor H. O. Otani, João Ricardo N. Vissoci, Lucas M. Marques, Ricardo R. Uchida.

**Validation:** Victor H. O. Otani, Felipe O. Aguiar, Hudson S. Buck, João Ricardo N. Vissoci, Lucas M. Marques, Ricardo R. Uchida.

**Visualization:** Victor H. O. Otani, Felipe O. Aguiar, Thiago P. Justino, Hudson S. Buck, Luiza B. Grilo, Matheus F. Figueiredo, Pedro M. Uchida, Daniel A. C. Vasques, Thaís Z. S. Otani, João Ricardo N. Vissoci, Lucas M. Marques, Ricardo R. Uchida.

**Writing – original draft:** Victor H. O. Otani, Felipe O. Aguiar, Thiago P. Justino, Hudson S. Buck, Luiza B. Grilo, Matheus F. Figueiredo, Pedro M. Uchida, João Ricardo N. Vissoci, Lucas M. Marques, Ricardo R. Uchida.

**Writing – review & editing:** Victor H. O. Otani, Felipe O. Aguiar, Thiago P. Justino, Hudson S. Buck, Luiza B. Grilo, Matheus F. Figueiredo, Pedro M. Uchida, Daniel A. C. Vasques, Thaís Z. S. Otani, João Ricardo N. Vissoci, Lucas M. Marques, Ricardo R. Uchida.

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
