## [Decision Letter · Decision Letter 0]

28 Jul 2025

PMEN-D-25-00224

ML-Based Detection of Depressive Profile Through Voice Analysis in WhatsApp™ Audio Messages of Brazilian Portuguese Speakers

PLOS Mental Health

Dear Dr. Marques,

Thank you for submitting your manuscript to PLOS Mental Health. After careful consideration, we feel that it has merit but does not fully meet PLOS Mental Health’s publication criteria as it currently stands. Therefore, we invite you to submit a revised version of the manuscript that addresses the points raised during the review process.

Consider all comments and concerns raised by reviewers.

We look forward to receiving your revised manuscript.

Kind regards,

Ariel Soares Teles

Academic Editor

PLOS Mental Health

Journal Requirements:

1. Please note that PLOS Mental Health has specific guidelines on code sharing for submissions in which author-generated code underpins the findings in the manuscript. In these cases, we expect all author-generated code to be made available without restrictions upon publication of the work. Please review our guidelines at https://journals.plos.org/mentalhealth/s/materials-and-software-sharing#loc-sharing-code and ensure that your code is shared in a way that follows best practice and facilitates reproducibility and reuse.

Additional Editor Comments (if provided):

Consider all comments and concerns raised by reviewers.

Reviewers' comments:

Reviewer's Responses to Questions

**Comments to the Author**

1. Does this manuscript meet PLOS Mental Health’s publication criteria?

Reviewer #1: Yes

Reviewer #2: Partly

2. Has the statistical analysis been performed appropriately and rigorously?

Reviewer #1: Yes

Reviewer #2: No

3. Have the authors made all data underlying the findings in their manuscript fully available (please refer to the Data Availability Statement at the start of the manuscript PDF file)?

Reviewer #1: Yes

Reviewer #2: Yes

4. Is the manuscript presented in an intelligible fashion and written in standard English?

Reviewer #1: Yes

Reviewer #2: Yes

Reviewer #1: Review Summary:

This study leverages WhatsApp voice messages from Brazilian Portuguese speakers to detect depression with traditional ML models. The real-world data scenario is commendable and the methodology is generally sound. However, key shortcomings such as sample imbalance, insufficient statistical testing, and limited interpretability must be thoroughly addressed before publication. And the combination of acoustic features and machine learning for depression detection has been extensively explored; this study does not adopt newer architectures (e.g., Transformers) nor does it benchmark results on additional datasets such as AVEC.

Required Revisions:

1. Increase the male sample size or apply resampling/re-weighting to mitigate gender imbalance.

2. Add confidence intervals and hypothesis tests (e.g., significance of AUC and accuracy differences).

3. Provide confusion matrices, FPR/FNR, and representative misclassification analyses to enhance transparency.

4. Recommend releasing anonymized code and acoustic-feature datasets to ensure reproducibility.

5. Recommend adding validation experiments on additional publicly available datasets (e.g., AVEC, DAIC-WOZ) to demonstrate cross-corpus generalizability and strengthen the robustness of the findings.

6. Discuss potential domain-shift effects between training (spontaneous speech) and testing (structured tasks).

7. Clarify how recording devices, ambient noise, and microphone variability were documented or controlled.

8. Explain the biological relevance of key features (e.g., MFCC2) to depressive pathophysiology.

9. Add sensitivity analyses or discussion on confounders such as medications.

10. Expand the ethics statement to detail voice de-identification, data storage, and access protocols.

11. Edit language for conciseness and unify terminology (e.g., “frame” vs. “sample”).

12. Refine Figures 3/4: clarify y-axis labels and legends to avoid ambiguity.

Reviewer #2: This manuscript presents a study on machine learning (ML)-based detection of depressive profiles using voice analysis from WhatsApp™ audio messages in Brazilian Portuguese speakers. The work addresses an important clinical need and demonstrates methodological rigor in several respects. However, there are a few areas where the statistical and methodological approach could be strengthened. Below are several suggestions to help improve the manuscript:

1. The study uses different speech types for training (spontaneous) and testing (structured), introducing a domain shift that may bias the model toward distinguishing speech styles rather than depressive states. It would enhance the study’s validity to evaluate performance within matched speech types, conduct ablation studies with mixed and separated datasets, and assess whether the key features identified (e.g., in Table 7) are consistent across validation schemes.

2. The reported model performance differs notably by gender (91.67% accuracy for women vs. 80% for men), but the male sample sizes are quite limited (8 in training, 16 in testing). It would be helpful to report confidence intervals, test the significance of these gender differences, and perform a power analysis. If the sample size is underpowered for subgroup analysis, the authors should consider reporting pooled results and clearly state the limitations.

3. The training and testing groups differ in diagnostic criteria: DSM-5-based clinical evaluations were used for the training set, while the MINI interview was used for the test set. This inconsistency raises concerns about the comparability and validity of the diagnostic labels. Clarifying these differences and discussing their implications would strengthen the clinical foundation of the study.

4. Table 7 shows considerable variation in the top 10 acoustic features across models, with only MFCC 2 appearing consistently. This variability raises concerns about feature stability and interpretability. It would be valuable to analyze whether the most important features remain consistent across validation folds and datasets.

5. While the paper highlights the practical potential of WhatsApp-based depression screening, several limitations remain underexplored. These include the study’s restriction to uncontrolled audio environments, and the observed gender disparities in performance. The authors are encouraged to discuss plans for handling diverse recording conditions, integrating the tool into real-world healthcare workflows, and addressing ethical considerations such as false positives, privacy, and informed consent.

**Do you want your identity to be public for this peer review?** For information about this choice, including consent withdrawal, please see our Privacy Policy

Reviewer #1: No

Reviewer #2: No

---

## [Decision Letter · Decision Letter 1]

9 Oct 2025

PMEN-D-25-00224R1

ML-Based Detection of Depressive Profile Through Voice Analysis in WhatsApp™ Audio Messages of Brazilian Portuguese Speakers

PLOS Mental Health

Dear Dr. Marques,

Thank you for submitting your manuscript to PLOS Mental Health. After careful consideration, we feel that it has merit but does not fully meet PLOS Mental Health’s publication criteria as it currently stands. Therefore, we invite you to submit a revised version of the manuscript that addresses the points raised during the review process.

We look forward to receiving your revised manuscript.

Kind regards,

Ariel Soares Teles

Academic Editor

PLOS Mental Health

Journal Requirements:

Additional Editor Comments:

Consider the comments from Reviewer #3.

Reviewers' comments:

Reviewer's Responses to Questions

**Comments to the Author**

Reviewer #2: All comments have been addressed

Reviewer #3: All comments have been addressed

publication criteria?

Reviewer #2: Yes

Reviewer #3: Partly

3. Has the statistical analysis been performed appropriately and rigorously?

Reviewer #2: Yes

Reviewer #3: Yes

4. Have the authors made all data underlying the findings in their manuscript fully available (please refer to the Data Availability Statement at the start of the manuscript PDF file)?

Reviewer #2: Yes

Reviewer #3: No

5. Is the manuscript presented in an intelligible fashion and written in standard English?

Reviewer #2: Yes

Reviewer #3: Yes

Reviewer #2: The authors have addressed all my concerns.

Reviewer #3: In this work, Authors evaluate several different ML models’ ability to classify depressed patients. The dataset is represented by several WhatsApp audio recordings, from which they extract numeorus audio features.

The problem of cheap and objective depression detection and evaluation tools is important nowadays, and the Machine Learning methodlogy seems robust.

However, I have some major concerns:

1. While detecting the presence of depression is surely an important task, as far as I know depression has several degrees. I wonder if reducing it to a dicotomic variable could be an oversimplification and, at the bottom line, present limited clinical benefits. Lines 69-71 seem to underline the same limitation: “[referred to tools used nowadays] Although these tools provide valuable insights into the presence and severity of depression, they often lack the specificity required for clinical phenotyping and individualized treatment strategies”. Does the models you provide have the specificity required for clinical phenotyping? Or are they to be intended as “pre-screening” tools?

2. Another major issue is the fact that performance on Test set for female subjects are sensibly higher than Train set and Test set for male subjects. For example, with kNN model:

1. Train set F1 = 75%, AUC = 69%

2. Test set on the past-week task, F1 = 84% female (+9% on train set) / 56% male (-19%), AUC = 84 female (+15%) / 60 male (-9%)

3. Test set on the count1-10 task, F1 = 81% female (+6% on train set) / 65% male (-10%), AUC = 83 female (+14%) / 65 male (-4%)

3. In my opinion, concerns presented at previous point could have several possible explanations:

1. Test set subjects were evaluated with MINI interview, differently from train subjects (lines 150-151: “structured psychiatric evaluation conducted by experienced psychiatrists in private clinical settings”);

2. Different content of speeches between Train and test set

3. Average ages look quite different between Train and Test set, expecially for female sample (36 for Train, 47 for Test). Have you evaluated this difference, other than the intra-group one (e.g., Test male MDD vs Test male HV)?

4. I would exclude data leakage since the ML methodology you provide seem robust

4. I would suggest to harmonize MDD evaluation and content of audio recordings to provide compatible datasets.

5. Still related to point 3, partial explanation of why performance for male sample is sensibly lower than femal sample is provided in lines 411-413. How would you motivate that? Are you suggesting that “traditional masculine norms” modify acoustic features as well? If the answer is yes, this is an interesting point of discussion. If the answer is no, I would expect an acoustic features-based model to be less sensible to such “norms”, as this could be a major limitation.

Some minor concerns:

• Line 262: please, indicate a reference for nested-CV.

• Figure 3 and 4 are hardly readable, please use high-resolution pictures.

• Line 356: what you developed are classifiers, not predictors.

• In Discussion, lines 366-376 are a repetition of what you described in Methods: please remove them.

• In Discussion, lines 452-473 present interesting data and facts. I would suggest to move them in the intro, to motivate the rationale of the work (see also comment 1).

• In Limitations, lines 490-495 describe a strength of the study. I would suggest to move in the Discussion.

• Table 6 and 8 report different statistical parameters, making comparison of results partially difficult. Please, be consistent in the presentation of results.

**Do you want your identity to be public for this peer review?** For information about this choice, including consent withdrawal, please see our Privacy Policy

Reviewer #2: No

Reviewer #3: No

---

## [Decision Letter · Decision Letter 2]

28 Oct 2025

PMEN-D-25-00224R2

ML-Based Detection of Depressive Profile Through Voice Analysis in WhatsApp™ Audio Messages of Brazilian Portuguese Speakers

PLOS Mental Health

Dear Dr. Marques,

Thank you for submitting your manuscript to PLOS Mental Health. After careful consideration, we feel that it has merit but does not fully meet PLOS Mental Health’s publication criteria as it currently stands. Therefore, we invite you to submit a revised version of the manuscript that addresses the points raised during the review process.

We look forward to receiving your revised manuscript.

Kind regards,

Ariel Soares Teles

Academic Editor

PLOS Mental Health

Journal Requirements:

Please note that PLOS Mental Health has specific guidelines on code sharing for submissions in which author-generated code underpins the findings in the manuscript. In these cases, we expect all author-generated code to be made available without restrictions upon publication of the work. Please review our guidelines at https://journals.plos.org/mentalhealth/s/materials-and-software-sharing#loc-sharing-code and ensure that your code is shared in a way that follows best practice and facilitates reproducibility and reuse.

Additional Editor Comments (if provided):

Consider the minor comment from Reviewer #3.

Reviewers' comments:

Reviewer's Responses to Questions

**Comments to the Author**

Reviewer #3: All comments have been addressed

publication criteria?

Reviewer #3: Yes

3. Has the statistical analysis been performed appropriately and rigorously?

Reviewer #3: Yes

4. Have the authors made all data underlying the findings in their manuscript fully available (please refer to the Data Availability Statement at the start of the manuscript PDF file)?

Reviewer #3: No

5. Is the manuscript presented in an intelligible fashion and written in standard English?

Reviewer #3: Yes

Reviewer #3: The quality of the work has surely improved. All my points have been addressed.

The only thing I ask is to add some reference to the claims related to the “masculine norms” (lines 497-501).

**Do you want your identity to be public for this peer review?** For information about this choice, including consent withdrawal, please see our Privacy Policy

Reviewer #3: No

---

## [Editor Report · Decision Letter 3]

3 Nov 2025

ML-Based Detection of Depressive Profile Through Voice Analysis in WhatsApp™ Audio Messages of Brazilian Portuguese Speakers

PMEN-D-25-00224R3

Dear Professor Marques,

We are pleased to inform you that your manuscript 'ML-Based Detection of Depressive Profile Through Voice Analysis in WhatsApp™ Audio Messages of Brazilian Portuguese Speakers' has been provisionally accepted for publication in PLOS Mental Health.

Best regards,

Ariel Soares Teles

Academic Editor

PLOS Mental Health